# Lower Serum Uric Acid Levels May Lower the Incidence of Diabetic Chronic Complications in U.S. Adults Aged 40 and Over

**DOI:** 10.3390/jcm12020725

**Published:** 2023-01-16

**Authors:** Yingdong Han, Shuolin Wang, He Zhao, Yu Cao, Xinxin Han, Hong Di, Yue Yin, Juan Wu, Yun Zhang, Xuejun Zeng

**Affiliations:** Department of Family Medicine & Division of General Internal Medicine, Department of Medicine, Chinese Academy of Medical Sciences, State Key Laboratory of Complex Severe and Rare Diseases (Peking Union Medical College Hospital), Peking Union Medical College Hospital, Beijing 100730, China

**Keywords:** uric acid, diabetic chronic complications, cardiovascular disease, diabetic kidney disease, diabetic peripheral neuropathy, diabetic retinopathy

## Abstract

Previous studies have generally reported the association between serum uric acid (SUA) and diabetic complications, but large-scale research exploring the above association in U.S. adults with diabetes is limited. To explore the association between SUA and chronic complications of diabetes among U.S. patients aged ≥40, we used data from the National Health and Nutrition Examination Survey 1999–2008. SUA was divided into three levels: T1 (SUA ≥ 420 μmol/L), T2 (300 ≤ SUA < 420 μmol/L), and T3 (SUA < 300 μmol/L). Binary logistic regression and restricted cubic spline analysis were applied to evaluate the association between SUA and chronic complications of diabetes. A trend test was performed as the SUA increased substantially. After full-adjusted confounding factors, patients in the T3 group had a lower risk of diabetic kidney disease, cardiovascular disease, and peripheral neuropathy compared with the T1 group, with a OR (95% CIs) of 0.33 (0.21–0.52), 0.56 (0.36–0.87), and 0.49 (0.27–0.89), respectively. The restricted cubic spline showed a significant positive relationship between SUA and cardiovascular disease and diabetic kidney disease in diabetes patients, but not peripheral neuropathy. Maintaining a SUA of less than 300 μmol/L might be protective against the risk of cardiovascular disease, diabetic kidney disease, and peripheral neuropathy other than diabetic retinopathy compared with a SUA of more than 420 μmol/L in U.S. diabetes patients aged 40 and over.

## 1. Introduction

Diabetes mellitus (DM), a complex and common metabolic disorder, occurs when the β cells of islets are unable to synthesize sufficient insulin or when the body is unable to use the insulin effectively. The prevalence of diabetes in adults has nearly doubled since 1980 (4.7% to 8.5%), with a steady increase globally, especially in middle-income countries [1,2]. Numerous previous studies have improved the understanding of pathogenesis of diabetes, whereas the prevalence of diabetes remains high due to changes in dietary patterns and related metabolic disorders, such as obesity and dyslipidemia. People with diabetes need continuous medical care and self-management education. Current treatments generally focus on slowing disease progression and preventing diabetic complications [3,4]. A chronic complication of DM is damage of blood vessels, due to long-term elevation of serum glucose levels, segmented under diabetic macrovascular disease and microvascular disease by vessel sizes [4]. Serum glucose variability and variability of blood pressure, obesity, and dyslipidemia are common risk factors for diabetic complications [5]. Previous studies suggested that uric acid (UA) might correlated with the development of diabetic complications, whereas the evidence was controversial [6,7,8,9]. Lu et al. found that SUA was associated with an increased prevalence of diabetic kidney disease (DKD) and cardiovascular disease (CVD) in men and postmenopausal women with DM [7].

Serum uric acid (SUA) is the ultimate product of purine metabolism [10]. Approximately two-thirds of UA is excreted by the kidneys and one-third is excreted by the gastrointestinal tract, and its synthesis and excretion are balanced under physiological conditions [11]. The onset of hyperuricemia is associated with overproduction or underexcretion of SUA [12]. Epidemiological studies suggested that the overall prevalence of hyperuricemia and gout has increased over the past decades [13,14]. Hyperuricemia is the precursor of gout, and apart from the involvement of joints, it is often accompanied by various comorbidities, including CVD, diabetes, chronic kidney disease, obesity, and dyslipidemia [10,15,16]. Hyperuricemia is positively correlated with the incidence of type 2 DM and the risk of DM reaching up to 27% in gout patients with SUA > 9 mg/dL [17,18]. The potential mechanism might be that hyperuricemia is correlated with increased insulin resistance and decreased release of insulin [18,19]. 

Previous clinical studies suggested that higher UA was correlated with a higher risk of CVD, DKD, and peripheral neuropathy (PN) in diabetes [6,7,8,9]. However, in their study of 4767 DM participants, Lu et al. found that SUA was not related to diabetic retinopathy (DR) in Chinese adults [7]. Previous research has generally reported the association between SUA and one certain type of diabetic complications, but large-scale research exploring the above association in U.S. adults with diabetes is limited. We performed this study with a nationally representative sample of U.S. DM patients whose data were obtained from the National Health and Nutrition Examination Survey (NHANES) 1999–2008 to investigate the relationship between SUA and diabetic chronic complications. We first explored the above relationship using NHANES data. Logistic regression analysis and restricted cubic spline analysis were used to detect the above relationship with a nationally representative sample.

## 2. Materials and Methods

### 2.1. Data Collection and Sample

NHANES collects the nutritional and health information of the U.S. population, which is conducted by the Centers for Disease Control and Prevention of America every 2 years. The survey consists of an in-home interview and physical examination, as well as urine and blood sample collection taken at a mobile examination center [20]. As the examinations on fundus and peripheral neuropathy were only performed in participants aged ≥40 years, our study only involved participants aged 40 and over. Our study was conducted following the Declaration of Helsinki [21]. 

### 2.2. Definitions and Measurement

Diabetes was defined as participants who met one of the followings: (1) self-reported diabetes; (2) participants with fasting glucose ≥7.0 mmol/L or HbA1C ≥ 6.5%; (3) currently receiving insulin or hypoglycemic drugs.

#### 2.2.1. Diabetic Kidney Disease

DKD was defined as diabetes with impaired glomerular filtration rate (GFR) and/or the presence of albuminuria. Albuminuria was defined as a ratio of urine albumin to creatinine (ACR) ≥ 30 mg/g. The estimation of GFR (eGFR) used the equation (eGFR = 141 × min (Scr/κ, 1)^α^ × max (Scr/κ, 1)^−1.209^ × 0.993^Age^ × 1.018 [if female] × 1.159 [if black]) and impaired GFR was defined as a GFR < 60 mL/min/1.73 m^2^ [22,23].

#### 2.2.2. Cardiovascular Disease

CVD was determined by a combination of standardized medical status questionnaires and self-reported physician diagnoses. Participants were asked the following 5 questions: (1) Have you ever been told that you have congestive heart failure? (2) Have you ever been told that you have coronary heart disease? (3) Have you ever been told that you have angina pectoris? (4) Have you ever been told that you have had a heart attack? (5) Have you ever been told that you have had a stroke?”. If the participant replied “yes” to any of the five questions, then he or she would be diagnosed with CVD [24]. 

#### 2.2.3. Peripheral Neuropathy

Participants lay on the examination table during the exam. Technicians applied slight pressure to the bottom of each of the participant’s feet at 3 sites with a standard monofilament: (1) plantar-first metatarsal head, (2) plantar-hallux, and (3) plantar-fifth metatarsal head. The above sites were tested in a non-sequential order by the examinee to allow for better discrimination of sensation, and then the number of insensate areas of each foot was calculated. Having at least one insensate site on either foot was diagnosed as PN, corresponding to a reduced sensation to touch [25]. 

#### 2.2.4. Diabetic Retinopathy

An ophthalmic digital imaging system was used to assess the presence of retinal diseases. Graders evaluated digital images at Wisconsin University. The levels of retinopathy were defined according to the grade protocol. A severity level ≥14 was defined as the presence of retinopathy, and a severity level of 10 to 13 was defined as the absence of retinopathy [26]. 

### 2.3. Inclusion and Exclusion Criteria

51,602 participants aged 18 and over were enrolled in the NHANES between 1999 and 2008. Exclusion criteria were as followed: (1) age < 40 year; (2) participants whose SUA data were not available; (3) self-reported malignancy disease; (4) being pregnant or breastfeeding; (5) participants without diabetes.

After screening complete data on urine ACR, serum creatinine, or standardized medical status questionnaires, 3075 and 3106 diabetic patients aged 40 and over were involved in the analysis of DKD and CVD, respectively. 

After screening complete data on peripheral neuropathy and fundus photography examinations, 1453 and 1233 diabetic patients aged 40 and over were involved in the analysis of PN and DR, respectively.

### 2.4. Variables

Variables in our study included age, sex, race, and education level. A waist circumference of more than 102 cm in men or more than 88 cm in women is defined as elevated waist circumference. Poverty income ratio (PIR) was used to define family income, with a PIR lower than 1.0 representing that live under the poverty line. This study also included total cholesterol, triglyceride, glycohemoglobin, and smoking status. The history of hypertension is defined as the self-reported diagnosis of hypertension by a doctor. The interval between the age at screening and the age at which they were first told they had diabetes was used to calculate the duration of diabetes (year). DM patients who were diagnosed less than 1 year were counted as half a year.

### 2.5. Statistical Analysis

All statistical analyses in our study were conducted with SPSS 23 and R 4.2.2. Non-normally distributed continuous variables were described with median (interquartile range). The Kruskal–Wallis test was used to compare the median among different SUA groups. The categorical variables were described with numbers (percentages) and the Chi-square test was used to compare the percentages among different SUA groups. The Bonferroni test was applied for the intergroup comparison. SUA in our study was divided into three levels: T1 (SUA ≥ 420 μmol/L), T2 (300 ≤ SUA < 420 μmol/L), and T3 (SUA < 300 μmol/L). T1 group was the reference group. Binary logistic regression explored the association between SUA and chronic complications of diabetes. Age, race, and gender were adjusted in Model 1, and Model 2 was additionally adjusted for hypertension, waist circumference, smoking, PIR, education, serum lipids, glycohemoglobin, and duration of diabetes. A trend test was performed as the SUA levels increased and logistic regression was performed to calculate the ORs with per SD increase in UA. We performed restricted cubic spline analysis with 3 knots of the SUA levels to characterize the dose–response relationship between SUA and diabetic complications in logistic regression Model 2. A two-sided *p* < 0.05 was considered statistically significant.

## 3. Results

### 3.1. Clinical Features of Included DM Patients

We summarized the clinical features of the involved diabetes patients for the analysis of cardiovascular disease (Table 1). The mean age was 64.12 ± 11.67 years and the mean duration was 12.87 ± 13.66 years. The mean eGFR of patients was 79.9 ± 25.03 mL/min/1.73 m^2^ and the mean SUA was 341.85 ± 96.26 μmol/L. A greater proportion of males belonged to the highest SUA levels (*p* < 0.01). With the increasing SUA levels, the median of HbA1c gradually decreased (*p* < 0.01), while the median of triglyceride, waist circumference, and creatinine gradually increased (*p* < 0.01). The proportion of smoking and prevalence of hypertension gradually increased with the increase of SUA levels (*p* < 0.01). The clinical characteristics of involved DM patients for the analysis of DKD, PN, and DR are presented in Appendix A. The percentage of patients with different diabetic chronic complications across different SUA levels are presented in Table 2. The proportion of DKD, CVD, and PN gradually increased with the increase of SUA levels (*p* < 0.05). 

The clinical features of DM patients with or without different diabetic chronic complications are shown in Appendix A. Patients with different complications had longer disease course and serum creatinine compared with patients without corresponding complication (*p* < 0.05). Diabetic complications accounted for greater proportion among patients receiving insulin therapy or hypertensive patients (*p* < 0.05).

### 3.2. Serum Uric Acid and Diabetic Kidney Disease

The crude ORs (95% CIs) of DKD were 0.48 (0.38–0.60) and 0.30 (0.23–0.40) in T2 and T3, respectively versus T1 group. In Model 1, the adjusted ORs (95% CIs) were 0.49 (0.39–0.63) and 0.34 (0.25–0.45), respectively, versus T1 group. In Model 2, the ORs (95% Cis) were 0.53 (0.38–0.74) and 0.33 (0.21–0.52), respectively, versus T1 of SUA. The trend test suggested that the ORs of DKD among different SUA groups gradually decreased (*p* for trend < 0.01) in all three models. Table 3 presented the results of binary logistic regression between SUA levels and the risk of DKD. Figure 1A presented the dose–response relationship between SUA and DKD. Uric acid was positively associated with the risk of DKD and presented a nonlinear dose–response relationship (*p* < 0.01, *p* for nonlinearity = 0.018). The reference point (OR = 1.0) of uric acid in the restricted cubic spline analysis was 334.41 μmol/L. 

### 3.3. Serum Uric Acid and Cardiovascular Disease

The ORs (95% CIs) of CVD were 0.54 (0.41–0.70) and 0.42 (0.32–0.56) in T2 and T3, respectively, versus the T1 group. In Model 2, the ORs (95% CIs) were 0.61 (0.41–0.90) and 0.56 (0.36–0.87), respectively, versus the T1 group. The trend test suggested that the ORs of CVD among different groups gradually decreased (*p* for trend < 0.05) in all three models. Detailed results are shown in Table 3. The dose–response relationship between uric acid and CVD is presented in Figure 1B. SUA levels were positively associated with the risk of CVD in diabetic patients aged 40 and over (*p* < 0.01, *p* for nonlinearity = 0.36). The reference point (OR = 1.0) of uric acid in the restricted cubic spline analysis was 332.41 μmol/L.

### 3.4. Serum Uric Acid and Peripheral Neuropathy

In Model 2, the multivariate-adjusted ORs (95% CIs) of PN were 0.59 (0.36–0.95) and 0.49 (0.27–0.89), respectively, versus T1 group. The trend test suggested that the ORs of PN among different SUA levels gradually decreased in crude model and full-adjusted Model 2 (*p* for trend < 0.05). Detailed results are shown in Table 3. Figure 1C presents dose–response relationship between SUA and PN. The reference point (OR = 1.0) of uric acid in the restricted cubic spline analysis was 327.27 μmol/L, and SUA was not significantly associated with the risk of PN among diabetic patients aged 40 and over (*p* = 0.20).

### 3.5. Serum Uric Acid and Diabetic Retinopathy

The relationship between SUA and DR and the trend test was insignificant in all three models (*p* for trend > 0.05). Detailed results are shown in Table 3. Figure 1D presented the dose–response relationship between SUA and DR. The reference point (OR = 1.0) of uric acid in the restricted cubic spline analysis was 333.69 μmol/L and uric acid was not significantly associated with the risk of DR in diabetic patients aged 40 and over (*p* = 0.83).

### 3.6. ORs of Diabetic Complications with per SD Increase of Uric Acid

We performed logistic regression to calculate the ORs per SD increase in UA, and the results are shown in Table 4. Uric acid was positively correlated with the risk of DKD and CVD after fully adjusting the confounding factors, and the ORs with per SD increase in SUA was 1.61 (1.35–1.92) and 1.21 (1.04–1.42). Uric acid was positively associated with the risk of PN in the crude model and the ORs with per SD increase in uric acid was 1.23 (1.06–1.43). However, SUA was not correlated with the risk of DR and PN in Model 2.

## 4. Discussion

Our study found that SUA levels ≤ 300 μmol/L might be a protective factor for the incidence of DKD, CVD, and PN compared with patients with a SUA level > 420 μmol/L. No correlation was observed between SUA and diabetic retinopathy. Uric acid was positively correlated with the risk of diabetic kidney disease and cardiovascular disease with per increase of SD. Restricted cubic spline revealed that SUA was positively associated with the risk of CVD and CVD.

Previous studies usually reported the relationship between SUA and one certain type of diabetic complication, and there were limited large-scale studies exploring the above association. We used a large nationally representative cohort among the U.S. population, fully adjusted for the potential covariates, and analyzed the relationship of SUA with different diabetic chronic complications at the same time, which increased the statistical strength and reliability of our results. This is the first study that used restricted cubic splines to explore the above relationship. Previous studies usually only reported the relationship between SUA and diabetic complications, and we have also provided specific targets of SUA management for DM patients. 

The pathophysiological mechanisms between SUA and chronic complications remain unclear. Potential mechanisms might be that increased SUA levels lead to the increase of ROS and promote the expression of inflammatory cytokine. For instance, IL-1β, IL-6, and TNF-α might lead to inflammation in vessels [27]. UA-mediated oxidative stress is associated with DNA damage, lipid peroxidation, and cellular damage [28]. UA is associated with endothelial dysfunction through inhibiting migration and proliferation of endothelial cells, NO bioavailability, and secretion in endothelial cells. Activation of RAAS induced high intraglomerular pressure, vascular dysfunction, and inflammation, and then led to cardiovascular and renal complications [27,29]. 

Lu et al. found a positive correlation between SUA levels and the prevalence of CVD and DKD, but not with that of DR in diabetic patients of Han Nationality [7], and a meta-analysis in 2016 found that hyperuricemia was correlated with an increased risk of PN [8]. Another study found that higher SUA concentration was associated with increased risk of developing eGFR decline among type 2 DM cases [9], which were consistent with our results. However, the first previous published study is limited to exploring the issue among Han nationality, which means it does not take into account other ethnic groups who might have different backgrounds. Additionally, the meta-analysis has failed to address the relationship between uric acid and various diabetic chronic complications. Based on the limitations of the previous research, our study makes an attempt to engage different ethnic groups and explore the relationship of SUA with different diabetic chronic complications. Furthermore, Liu et al. found that elevated uric acid levels were associated with a higher risk of increase in severity of DR over a 3-year follow-up [30]. Hayashino et al. found that both lower and higher uric acid levels were independently correlated with the risk of progression in albuminuria other than the development of albuminuria [31], and Cai’s newly published research suggested that low SUA level was closely correlated with PN [32]. These findings were inconsistent with our study. However, Liu’s study was a hospital-based but not a community-based study. Hayashino’s study used single urine ACR measurements but failed to combine GFR for the evaluation of DKD and Cai’s study was limited to type 2 DM patients without hyperuricemia. We used a large representative nationwide sample, and GFR was also considered in the definition of DKD. Our study makes attempts to involve DM patients with different SUA levels, which makes our results more convincing. In addition, the statistical analysis, study population, and adjusted covariates differed in the above-mentioned research. For instance, age, sex, eGFR, HbA1C, and duration of DM were adjusted in Seok Kang’s research [33], while systolic/diastolic blood pressure, serum lipid, and waist circumference were adjusted in Liu’s study [30], and our study additionally adjusted race, education levels, and poverty income ratio, which might lead to the different results. Participants involved in the above-mentioned research had different characteristics. Lu’s published study was limited to men aged ≥18 years and postmenopausal women [7]. The prevalence of chronic complications was lower in younger DM patients, which might hinder the discovery of a significant correlation. Our study involved DM patients aged 40 and over whose prevalence of diabetic complications was higher compared with younger patients. The age difference of the involved participants means the different proportions of type 1 and type 2 DM in the study population, which might also influence the results. 

We found a significant correlation between SUA and PN in binary logistic regression analysis, but it became insignificant in restricted cubic spline analysis. Different references used in the above analysis might be potential explanations. Binary logistic regression analysis took patients with SUA > 420 μmol/L as a reference, while the reference point of uric acid in the restricted cubic spline was 327.27 μmol/L. In addition, SUA was transformed into an ordered variable in logistic regression analysis, while SUA was a continuous variable in restricted cubic spline analysis. It did not make sense that SUA levels were negatively correlated with HbA1c in our study, but it was consistent with some previous studies [7,31,34]. More patients in the higher SUA group receiving insulin therapy might explain this phenomenon. 

The limitations of our study are as follows. Primarily, it was hard to determine causal associations, as it was a cross-sectional study. Further mendelian randomization studies, basic research, and large-scale cohort studies are required to confirm the causality. Due to the limited availability of data, we applied data from NHANES 1999 to 2008. It was difficult to include all potential covariates, some of which might be considered outdated. SUA in our study was only measured once while UA was a dynamic variable, which might be affected by the diet and cause some bias. Due to the limitation of data on oral glucose tolerance tests, some patients with diabetes might not be included. Finally, keeping a common quartile range for both genders may cause some bias.

The prevalence of hyperuricemia is gradually increasing with improved living conditions and changes in dietary patterns. It is controversial whether to initiate urate-lowering therapy (ULT) for asymptomatic hyperuricemia patients. The 2020 American College of Rheumatology Guideline recommends against initiating any pharmacologic ULT in individuals with asymptomatic hyperuricemia [35], while the Japanese Society of Gout recommends initiating drug therapy when the SUA reaches 8.0 mg/dL or more [36]. From our study, more attention should be paid to the SUA levels for patients with DM and maintaining a SUA level of less than 300 μmol/L might be protective against the risk of diabetic chronic complications.

## 5. Conclusions

We found that maintaining SUA levels at less than 300 μmol/L might be protective against the risk of DKD, CVD, and PN other than DR compared with patients with SUA values more than 420 μmol/L. We hope our study can provide valuable evidence for the management of diabetic patients with hyperuricemia. Future prospective studies are necessary to confirm the relationship between SUA and diabetic chronic complications and the protective effect of urate-lowering therapy.

## Figures and Tables

**Figure 1 jcm-12-00725-f001:**
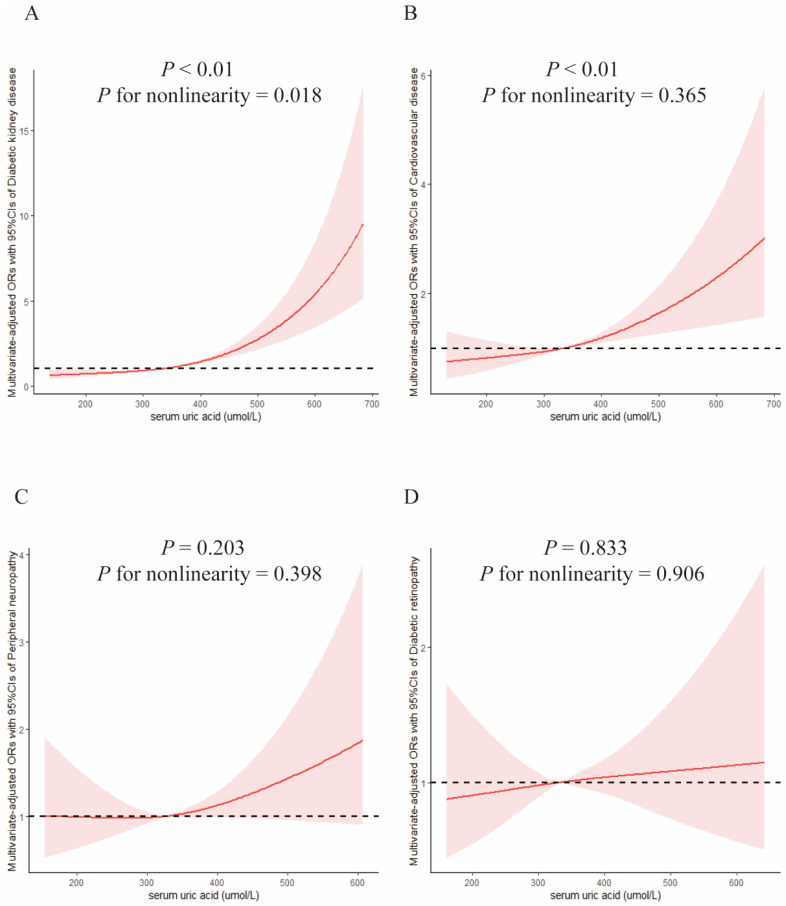
Examination of the dose–response relationship between serum uric acid (μmol/L) and the risk of diabetic chronic complications by restricted cubic splines model. The restricted cubic splines model adjusted for age, gender, race, poverty–income ratio, waist circumference, smoking status, education level, hypertension, serum triglyceride, total cholesterol, courses of diabetes, and HbA1c. (**A**) Diabetic kidney disease; (**B**) Cardiovascular disease; (**C**) Peripheral neuropathy; (**D**) Diabetic retinopathy.

**Table 1 jcm-12-00725-t001:** Clinical characteristics of the involved diabetes patients for the analysis of cardiovascular disease (*n* = 3106).

SUA Levels	SUA ≤ 300	300 < SUA ≤ 420	SUA > 420	*p* Value
Age (year) ^†^	62 (19)	65 (17)	67 (16)	<0.01
Males (%) ^‡^	492 (43.3)	726 (53.6)	387 (62.9)	<0.01
Race (%) ^‡^				<0.01
Mexican American	367 (32.3)	321 (23.7)	60 (9.8)	
Other Hispanic	83 (7.3)	75 (5.5)	25 (4.1)	
Non-Hispanic White	417 (36.7)	580 (42.8)	285 (46.3)	
Non-Hispanic Black	225 (19.8)	340 (25.1)	221 (35.9)	
Other race	45 (4.0)	38 (2.8)	24 (3.9)	
Education level (%) ^‡^				<0.01
Less than 9th grade	331 (29.1)	358 (26.4)	109 (17.7)	
9–11th grade	243 (21.4)	234 (17.3)	131 (21.3)	
High school graduate	222 (19.5)	304 (22.5)	165 (26.8)	
College or AA degree	238 (20.9)	284 (21.0)	131 (21.3)	
College graduate or above	103 (9.1)	173 (12.7)	79 (12.8)	
Waist circumference (cm) ^†^	101.9 (17.5)	107.6 (19.3)	110.3 (19.5)	<0.01
Cholesterol (mmol/L) ^†^	5.04 (1.47)	4.91 (1.50)	4.89 (1.53)	0.178
Triglyceride (mmol/L) ^†^	1.72 (1.37)	1.82 (1.55)	1.91 (1.48)	<0.01
Creatinine (μmol/L) ^†^	70.7 (26.52)	79.6 (26.54)	99.0 (51.27)	<0.01
Poverty income ratio < 1 (%) ^‡^	238 (22.9)	275 (22.2)	98 (17.7)	0.042
Serum uric acid (μmol/L) ^†^	255.8 (53.6)	350.9 (53.6)	469.9 (71.3)	<0.01
Glycohemoglobin (%) ^†^	7.2 (2.4)	6.7 (1.5)	6.6 (1.3)	<0.01
Taking insulin now (%) ^‡^	208 (18.5)	221 (16.5)	130 (21.3)	0.074
Fasting glucose (mmol/L) ^†^	7.88 (4.94)	7.22 (3.27)	7.22 (2.77)	<0.01
Duration of diabetes (year) ^†^	9 (13)	8 (14)	10 (15)	0.034
Hypertension (%) ^‡^	602 (52.9)	911 (67.3)	481 (78.2)	<0.01
Smoked at least 100 cigarettes in life (%) ^‡^	568 (50.0)	733 (54.1)	366 (59.5)	<0.01

SUA: Serum uric acid. Data are number of subjects (percentage) or medians (interquartile ranges). ^†^ The Kruskal–Wallis test was used to compare the median values among participants in different groups. ^‡^ Chi-square test was used to compare the percentage among participants in different groups.

**Table 2 jcm-12-00725-t002:** The number (percentage) of patients with different chronic complications of diabetes across different serum uric acid levels.

SUA Levels	SUA ≤ 300	300 < SUA ≤ 420	SUA > 420	*p* Value
Diabetic kidney disease (%) ^‡^	397 (35.6)	621 (46.0)	380 (62.4)	<0.01
Cardiovascular disease (%) ^‡^	252 (22.2)	385 (28.4)	252 (41.0)	<0.01
Peripheral neuropathy (%) ^‡^	140 (25.0)	179 (27.8)	84 (33.6)	0.041
Diabetic retinopathy (%) ^‡^	148 (35.0)	186 (34.1)	93 (35.1)	0.947

SUA: Serum uric acid. Data are number of subjects (percentage). ^‡^ Chi-square test was used to compare the percentage among participants in different groups.

**Table 3 jcm-12-00725-t003:** Weighted odds ratios (95% confidence intervals) for chronic complication of people with diabetes across different levels of serum uric acid (SUA) (*n* = 3106).

	Case/Participants	Crude ^†^	Model 1 ^†^	Model 2 ^†^
Cardiovascular disease				
SUA > 420	615/3106	1.00 (Ref.)	1.00 (Ref.)	1.00 (Ref.)
300 < SUA ≤ 420	1354/3106	0.54 (0.41–0.70) **	0.58 (0.44–0.77) **	0.61 (0.41–0.90) *
SUA ≤ 300	1137/3106	0.42 (0.32–0.56) **	0.54 (0.39–0.73) **	0.56 (0.36–0.87) *
*p* for trend		<0.01	<0.01	<0.05
Diabetic kidney disease				
SUA > 420	609/3075	1.00 (Ref.)	1.00 (Ref.)	1.00 (Ref.)
300 < SUA ≤ 420	1350/3075	0.48 (0.38–0.60) **	0.49 (0.39–0.63) **	0.53 (0.38–0.74) **
SUA ≤ 300	1116/3075	0.30 (0.23–0.40) **	0.34 (0.25–0.45) **	0.33 (0.21–0.52) **
*p* for trend		<0.01	<0.01	<0.01
Diabetic peripheral neuropathy				
SUA > 420	250/1453	1.00 (Ref.)	1.00 (Ref.)	1.00 (Ref.)
300 < SUA ≤ 420	643/1453	0.59 (0.42–0.84) **	0.65 (0.45–0.94) *	0.59 (0.36–0.95) *
SUA ≤ 300	560/1453	0.49 (0.33–0.73) **	0.62 (0.40–0.97) *	0.49 (0.27–0.89) *
*p* for trend		<0.01	0.07	<0.05
Diabetic retinopathy				
SUA > 420	265/1233	1.00 (Ref.)	1.00 (Ref.)	1.00 (Ref)
300 < SUA ≤ 420	545/1233	0.89 (0.56–1.44)	1.00 (0.61–1.64)	1.01 (0.54–1.91)
SUA ≤ 300	423/1233	0.97 (0.66–1.43)	1.20 (0.77–1.86)	0.89 (0.53–1.50)
*p* for trend		0.99	0.35	0.62

SUA: Serum uric acid. ^†^ Calculated using binary logistic regression. Model 1 adjusted for age, gender, and race. Model 2 adjusted for poverty–income ratio, waist circumference, smoking status, education level, hypertension, serum triglyceride, total cholesterol, courses of diabetes (year), and HbA1c. * *p* < 0.05. ** *p* < 0.01.

**Table 4 jcm-12-00725-t004:** Weighted odds ratios (95% confidence intervals) for chronic complications of people with diabetes with the increase per standard deviation uric acid.

	Crude ^†^	Model 1 ^†^	Model 2 ^†^
Diabetic kidney disease	1.64 (1.49–1.82) **	1.59 (1.44–1.75) **	1.61 (1.35–1.92) **
Cardiovascular disease	1.41 (1.27–1.56) **	1.29 (1.16–1.44) **	1.21 (1.04–1.42) *
Peripheral neuropathy	1.23 (1.06–1.43) **	1.19 (0.95–1.32)	1.23 (0.97–1.55)
Diabetic retinopathy	0.99 (0.85–1.15)	0.92 (0.77–1.10)	0.99 (0.81–1.22)

^†^ Calculated using binary logistic regression. Model 1 adjusted for age, gender and race. Model 2 adjusted for poverty–income ratio, waist circumference, smoking status, education level, hypertension, serum triglyceride, total cholesterol, courses of diabetes (year), HbA1c. * *p* < 0.05. ** *p* < 0.01.

## Data Availability

Datasets that support the conclusions of our research can be found in the public repository, as described below. The authors do not own the data. The Data are available from the National Center for Health Statistics (http://www.cdc.gov/nchs/nhanes/nhanes_questionnaires.htm, accessed on 15 August 2022).

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
