# Peer review of "Lower Serum Uric Acid Levels May Lower the Incidence of Diabetic Chronic Complications in U.S. Adults Aged 40 and Over"

_jcm, 2023, doi:10.3390/jcm12020725_

Round 1
Reviewer 1 Report
Comments
The manuscript presents an interesting analysis of the relationship between uric acid and diabetic complications using NHANES data. This manuscript will need some language editing.
· Please add to the introduction the recent studies from NHANES data about the relationship between serum uric acid and diabetic complications, if there is no such study please mention that.
· Gender is known to affect the relationship between serum uric acid level and diabetic complications (Wang, J, Yu, Y, Li, X, et al. Diabetes Metab Res Rev. 2018; 34:e3046. https://doi.org/10.1002/dmrr.3046). The normal range of serum uric acid also differs between men and women. Therefore, keeping a common quartile range for the study population may lead to biased estimation. The authors may explain the rationale behind keeping a common range for uric acid for both genders.
Author Response
Dear Editor and reviewer,
Thanks very much for consideration of our manuscript entitled " Lower serum uric acid levels may lower the incidence of diabetic chronic complications in U.S. adults aged 40 and over". We have modified the manuscript according to the comments, and all the changes to the manuscript have been highlighted in the text (with yellow highlighted text). The responses have been given in the following.
We look forward to your response.
Thank you and best regards.
Sincerely yours,
Yingdong Han
Hanyingdong_hyd@126.com
Pro. Yun Zhang and Xuejun Zeng
zhangyun10806@pumch.cn; zxjpumch@126.com.
Reviewer #1 Comments:
- The manuscript presents an interesting analysis of the relationship between uric acid and diabetic complications using NHANES data. This manuscript will need some language editing.
Reply: Thanks for your suggestion. We have asked another native English speaker to check and revise the grammar and polish our article.
- Please add to the introduction the recent studies from NHANES data about the relationship between serum uric acid and diabetic complications, if there is no such study, please mention that.
Reply: Thanks for your suggestion. Previous studies have generally reported the association between SUA and one certain type of diabetic complications, but large-scale researches exploring the above association in U.S. adults with diabetes are limited. We firstly explored above relationship using NHANES data. We performed this study with a nationally representative cohort of U.S. adults with diabetes whose data were obtained from the National Health and Nutrition Examination Survey (NHANES) 1999-2008 to explore the association between SUA and diabetic chronic complications. (Introduction section, Line 64-69)
- Gender is known to affect the relationship between serum uric acid level and diabetic complications (Wang, J, Yu, Y, Li, X, et al. Diabetes Metab Res Rev. 2018; 34:e3046. https://doi.org/10.1002/dmrr.3046). The normal range of serum uric acid also differs between men and women. Therefore, keeping a common quartile range for the study population may lead to biased estimation.The authors may explain the rationale behind keeping a common range for uric acid for both genders.
Reply: Thanks for your suggestion. We appreciated the thoughtful comments and suggestion. The normal range of serum uric acid differs between men and women.
We divided the participants into 3 groups according to 3 levels of serum uric acid concentration, which is helpful to detect the real effect of different uric acid concentration on diabetic complications. For the first time, we found that maintaining SUA levels less than 300μmol/L might be protective against the risk of DKD, CVD and PN compared with patients with SUA values more than 420umol/L, which might provide valuable evidence for the management of diabetic patients with hyperuricemia.
In addition, our study only involved participants aged 40 and over and most of the women in our sample are postmenopausal. Menopause is independently associated with higher serum uric acid levels [1]. So, we keep a common range for uric acid for both genders in this study.
We agree with your idea. Keeping a common quartile range for the study population may lead to biased estimation. we have mentioned it in the limitations section (Discussion section, Line 326-327).
Ref:
[1] Hak AE, Choi HK. Menopause, postmenopausal hormone use and serum uric acid levels in US women--the Third National Health and Nutrition Examination Survey. Arthritis Res Ther. 2008;10(5):R116. doi: 10.1186/ar2519.
Reviewer 2 Report
Dear author, Congratulation for your work.
Authors have studied to evaluate the Serum uric acid association in respect to diabetic complication. Manuscript was well written, although following comments needs to be addressed.
Abstract: Kindly start with basic information in first statement. then switch to the main aim and objective.
Introduction: redraft uric acid pathophysiology in a couple of statement line no: 44 to 50. Provide some key references on SUA relation with the diabetic complication.
Method: permission to use data of CDC was taken or not?
Discussion: Provide some key references.
Overall, manuscript was well, written. Conclusion is consistent with the evidence presented. Images are clear.
Author Response
Dear Editor and reviewer,
Thanks very much for consideration of our manuscript entitled " Lower serum uric acid levels may lower the incidence of diabetic chronic complications in U.S. adults aged 40 and over". We have modified the manuscript according to the comments, and all the changes to the manuscript have been highlighted in the text (with yellow highlighted text). The responses have been given in the following.
We look forward to your response.
Thank you and best regards.
Sincerely yours,
Yingdong Han
Hanyingdong_hyd@126.com
Pro. Yun Zhang and Xuejun Zeng
zhangyun10806@pumch.cn; zxjpumch@126.com.
Reviewer #2 Comments:
Authors have studied to evaluate the Serum uric acid association in respect to diabetic complication. Manuscript was well written, although following comments needs to be addressed.
1.Abstract: Kindly start with basic information in first statement. then switch to the main aim and objective.
Reply: Thanks for your suggestion. We have added the basic information in the Abstract section (Abstract section, Line 10-13).
Previous studies have generally reported the association between serum uric acid (SUA) and diabetic complications, but large-scale researches exploring the above association in U.S. adults with diabetes are limited. To determine the association between SUA and chronic complications of diabetes among U.S. patients aged 40 and over.
2.Introduction: redraft uric acid pathophysiology in a couple of statement line no: 44 to 50. Provide some key references on SUA relation with the diabetic complication.
Reply: Thanks for your suggestion. We have redrafted the pathophysiology of uric acid in introduction section (Introduction section, Line 48-51).
We have provided some key references on SUA relation with the diabetic complication in text, reference [6-9] (Introduction section, Line 43-47 and Line 60-63).
Ref:
- Verma S, Ji Q, Bhatt DL, et al. Association between uric acid levels and cardio-renal outcomes and death in patients with type 2 diabetes: A subanalysis of EMPA-REG OUTCOME. Diabetes, obesity & metabolism 22(7), 1207-14 (2020). doi:10.1111/dom.13991.
- Wan H, Wang Y, Chen Y, et al. Different associations between serum urate and diabetic complications in men and postmenopausal women. Diabetes Res Clin Pract 160, 108005 (2020). doi:10.1016/j.diabres.2020.108005.
- Yu S, Chen Y, Hou X, et al. Serum Uric Acid Levels and Diabetic Peripheral Neuropathy in Type 2 Diabetes: a Systematic Review and Meta-analysis. Mol Neurobiol 53(2), 1045-51 (2016). doi:10.1007/s12035-014-9075-0.
- Wang J, Yu Y, Li X, et al. Serum uric acid levels and decreased estimated glomerular filtration rate in patients with type 2 diabetes: A cohort study and meta-analysis. Diabetes Metab Res Rev 34(7), e3046 (2018). doi:10.1002/dmrr.3046.
3.Method: permission to use data of CDC was taken or not?
Reply: Thanks for your suggestion. The datasets supporting the conclusions of this article are available in the public repository. National Health and Nutrition Examination Survey data are available from the National Center for Health Statistics (http://www.cdc.gov/nchs/nhanes/nhanes_questionnaires.htm).
So, no permission is required to use the NHANES data.
- Discussion: Provide some key references.
Reply: Thanks for your suggestion. We have provided some key references in the discussion section. These studies were of high quality.
Ref:
- Verma S, Ji Q, Bhatt DL, et al. Association between uric acid levels and cardio-renal outcomes and death in patients with type 2 diabetes: A subanalysis of EMPA-REG OUTCOME. Diabetes, obesity & metabolism 22(7), 1207-14 (2020). doi:10.1111/dom.13991.
- Wan H, Wang Y, Chen Y, et al. Different associations between serum urate and diabetic complications in men and postmenopausal women. Diabetes Res Clin Pract 160, 108005 (2020). doi:10.1016/j.diabres.2020.108005.
- Yu S, Chen Y, Hou X, et al. Serum Uric Acid Levels and Diabetic Peripheral Neuropathy in Type 2 Diabetes: a Systematic Review and Meta-analysis. Mol Neurobiol 53(2), 1045-51 (2016). doi:10.1007/s12035-014-9075-0.
- Wang J, Yu Y, Li X, et al. Serum uric acid levels and decreased estimated glomerular filtration rate in patients with type 2 diabetes: A cohort study and meta-analysis. Diabetes Metab Res Rev 34(7), e3046 (2018). doi:10.1002/dmrr.3046.
- Lee JJ, Yang IH, Kuo HK, et al. Serum uric acid concentration is associated with worsening in severity of diabetic retinopa-thy among type 2 diabetic patients in Taiwan--a 3-year prospective study. Diabetes Res Clin Pract 106(2), 366-72 (2014). doi:10.1016/j.diabres.2014.07.027.
- Hayashino Y, Okamura S, Tsujii S, Ishii H. Association of serum uric acid levels with the risk of development or progression of albuminuria among Japanese patients with type 2 diabetes: a prospective cohort study [Diabetes Distress and Care Registry at Tenri (DDCRT 10)]. Acta Diabetol 53(4), 599-607 (2016). doi:10.1007/s00592-015-0825-x.
- Zhuang Y, Huang H, Hu X, Zhang J, Cai Q. Serum uric acid and diabetic peripheral neuropathy: a double-edged sword. Acta Neurol Belg (2022). doi:10.1007/s13760-022-01978-1.
- Kim HK, Lee M, Lee YH, et al. Uric Acid Variability as a Predictive Marker of Newly Developed Cardiovascular Events in Type 2 Diabetes. Front Cardiovasc Med 8, 775753 (2021). doi:10.3389/fcvm.2021.775753.
Reviewer 3 Report
Lower serum uric acid levels may lower the incidence of dia- 2 betic chronic complications in U.S. adults aged 40 and over. The subject developed in the article is welcome for patients with diabetes. The 7 tables are carefully structured. The first 3 show the clinical characteristics of diabetic patients with renal complications, peripheral neuropathy and retinopathy. The following 4 tables show diabetic patients with diabetic complications compared to controls without diabetes. In addition, a classification of patients is made according to 3 levels of serum uric acid concentration, , a very important fact to establish the protective/damaging effect of uric acid.
Author Response
Dear Editor and reviewer,
Thanks very much for consideration of our manuscript entitled " Lower serum uric acid levels may lower the incidence of diabetic chronic complications in U.S. adults aged 40 and over". We have modified the manuscript according to the comments, and all the changes to the manuscript have been highlighted in the text (with yellow highlighted text). The responses have been given in the following.
We look forward to your response.
Thank you and best regards.
Sincerely yours,
Yingdong Han
Hanyingdong_hyd@126.com
Pro. Yun Zhang and Xuejun Zeng
zhangyun10806@pumch.cn; zxjpumch@126.com.
Reviewer #3 Comments:
- Lower serum uric acid levels may lower the incidence of diabetic chronic complications in U.S. adults aged 40 and over. The subject developed in the article is welcome for patients with diabetes. The 7 tables are carefully structured. The first 3 show the clinical characteristics of diabetic patients with renal complications, peripheral neuropathy and retinopathy. The following 4 tables show diabetic patients with diabetic complications compared to controls without diabetes. In addition, a classification of patients is made according to 3 levels of serum uric acid concentration, a very important fact to establish the protective/damaging effect of uric acid.
Reply: Thank you very much.